# Sharpness, Restart and Acceleration

**Vincent Roulet**
INRIA, ENS
Paris France
vincent.roulet@inria.fr

**Alexandre d'Aspremont**
CNRS, ENS
Paris France
aspremon@ens.fr

## Abstract

The Łojasiewicz inequality shows that sharpness bounds on the minimum of convex optimization problems hold almost generically. Sharpness directly controls the performance of restart schemes, as observed by Nemirovskii and Nesterov [1985]. The constants quantifying error bounds are of course unobservable, but we show that optimal restart strategies are robust, and searching for the best scheme only increases the complexity by a logarithmic factor compared to the optimal bound. Overall then, restart schemes generically accelerate accelerated methods.

## Introduction

We study convex optimization problems of the form

$$\text{minimize} \quad f(x) \tag{P}$$

where $f$ is a convex function defined on $\mathbb{R}^n$. The complexity of these problems using first order methods is generically controlled by smoothness assumptions on $f$ such as Lipschitz continuity of its gradient. Additional assumptions such as strong convexity or uniform convexity provide respectively linear [Nesterov, 2013b] and faster polynomial [Juditski and Nesterov, 2014] rates of convergence. However, these assumptions are often too restrictive to be applied. Here, we make a much weaker and generic assumption that describes the sharpness of the function around its minimizers by constants $\mu \geq 0$ and $r \geq 1$ such that

$$\frac{\mu}{r}d(x, X^*)^r \leq f(x) - f^*, \quad \text{for every } x \in K, \tag{Sharp}$$

where $f^*$ is the minimum of $f$, $K \subset \mathbb{R}^n$ is a compact set, $d(x, X^*) = \min_{y \in X^*} \|x - y\|$ is the distance from $x$ to the set $X^* \subset K$ of minimizers of $f$[1] for the Euclidean norm $\|\cdot\|$. This defines a *lower bound* on the function around its minimizers: for $r = 1$, $f$ shows a kink around its minimizers and the larger is $r$ the flatter is the function around its minimizers. We tackle this property by restart schemes of classical convex optimization algorithms.

Sharpness assumption (Sharp) is better known as a Hölderian error bound on the distance to the set of minimizers. Hoffman [Hoffman, 1952] first introduced error bounds to study system of linear inequalities. Natural extensions were then developed for convex optimization [Robinson, 1975; Mangasarian, 1985; Auslender and Crouzeix, 1988], notably through the concept of sharp minima [Polyak, 1979; Burke and Ferris, 1993; Burke and Deng, 2002]. But the most striking discovery was made by Łojasiewicz [Łojasiewicz, 1963, 1993] who proved inequality (Sharp) for real analytic and subanalytic functions. It has then been extended to non-smooth subanalytic convex functions by Bolte et al. [2007]. Overall, since (Sharp) essentially measures the sharpness of minimizers, it holds somewhat generically. On the other hand, this inequality is purely descriptive as we have no hope of ever observing either $r$ or $\mu$, and deriving adaptive schemes is crucial to ensure practical relevance.

Łojasiewicz inequalities either in the form of (Sharp) or as gradient dominated properties [Polyak, 1979] led to new simple convergence results [Karimi et al., 2016], in particular for alternating and splitting methods [Attouch et al., 2010; Frankel et al., 2015], even in the non-convex case [Bolte et al., 2014]. Here we focus on Hölderian error bounds as they offer simple explanation of accelerated rates of restart schemes.

Restart schemes were already studied for strongly or uniformly convex functions [Nemirovskii and Nesterov, 1985; Nesterov, 2013a; Juditski and Nesterov, 2014; Lin and Xiao, 2014]. In particular, Nemirovskii and Nesterov [1985] link a "strict minimum" condition akin to (Sharp) with faster convergence rates using restart schemes which form the basis of our results, but do not study the cost of adaptation and do not tackle the non-smooth case. In a similar spirit, weaker versions of this strict minimum condition were used more recently to study the performance of restart schemes in [Renegar, 2014; Freund and Lu, 2015; Roulet et al., 2015]. The fundamental question of a restart scheme is naturally to know when must an algorithm be stopped and relaunched. Several heuristics [O'Donoghue and Candes, 2015; Su et al., 2014; Giselsson and Boyd, 2014] studied adaptive restart schemes to speed up convergence of optimal methods. The robustness of restart schemes was then theoretically studied by Fercoq and Qu [2016] for quadratic error bounds, i.e. (Sharp) with $r = 2$, that LASSO problem satisfies for example. Fercoq and Qu [2017] extended recently their work to produce adaptive restarts with theoretical guarantees of optimal performance, still for quadratic error bounds. Previous references focus on smooth problems, but error bounds appear also for non-smooth ones, Gilpin et al. [2012] prove for example linear converge of restart schemes in bilinear matrix games where the minimum is sharp, i.e. (Sharp) with $r = 1$.

Our contribution here is to derive optimal scheduled restart schemes for general convex optimization problems for smooth, non-smooth or Hölder smooth functions satisfying the sharpness assumption. We then show that for smooth functions these schemes can be made adaptive with nearly optimal complexity (up to a squared log term) for a wide array of sharpness assumptions. We also analyze restart criterion based on a sufficient decrease of the gap to the minimum value of the problem, when this latter is known in advance. In that case, restart schemes are shown ot be optimal without requiring any additional information on the function.

# 1 Problem assumptions

## 1.1 Smoothness

Convex optimization problems (P) are generally divided in two classes: smooth problems, for which $f$ has Lipschitz continuous gradients, and non-smooth problems for which $f$ is not differentiable. Nesterov [2015] proposed to unify point of views by assuming generally that there exist constants $1 \leq s \leq 2$ and $L > 0$ such that

$$\|\nabla f(x) - \nabla f(y)\| \leq L\|x - y\|^{s-1}, \quad \text{for all } x, y \in \mathbb{R}^n \qquad \text{(Smooth)}$$

where $\nabla f(x)$ is any sub-gradient of $f$ at $x$ if $s = 1$ (otherwise this implies differentiability of $f$). For $s = 2$, we retrieve the classical definition of smoothness [Nesterov, 2013b]. For $s = 1$ we get a classical assumption made in non-smooth convex optimization, i.e., that sub-gradients of the function are bounded. For $1 < s < 2$, this assumes gradient of $f$ to be Hölder Lipschitz continuous. In a first step, we will analyze restart schemes for smooth convex optimization problems, then generalize to general smoothness assumption (Smooth) using appropriate accelerated algorithms developed by Nesterov [2015].

## 1.2 Error bounds

In general, an error bound is an inequality of the form

$$d(x, X^*) \leq \omega(f(x) - f^*),$$

where $\omega$ is an increasing function at 0, called the residual function, and $x$ may evolve either in the whole space or in a bounded set, see Bolte et al. [2015] for more details. We focus on Hölderian Error Bounds (Sharp) as they are the most common in practice. They are notably satisfied by a analytic and subanalytic functions but the proof (see e.g. Bierstone and Milman [1988, Theorem 6.4]) is shown using topological arguments that are far from constructive. Hence, outside of some

particular cases (e.g. strong convexity), we cannot assume that the constants in (Sharp) are known, even approximately.

Error bounds can generically be linked to Łojasiewicz inequality that upper bounds magnitude of the gradient by values of the function [Bolte et al., 2015]. Such property paved the way to many recent results in optimization [Attouch et al., 2010; Frankel et al., 2015; Bolte et al., 2014]. Here we will see that (Sharp) is sufficient to acceleration of convex optimization algorithms by their restart. Note finally that in most cases, error bounds are *local properties* hence the convergence results that follow will generally be local.

### 1.3 Sharpness and smoothness

Let $f$ be a convex function on $\mathbb{R}^n$ satisfying (Smooth) with parameters $(s, L)$. This property ensures that, $f(x) \leq f^* + \frac{L}{s} \|x - y\|^s$, for given $x \in \mathbb{R}^n$ and $y \in X^*$. Setting $y$ to be the projection of $x$ onto $X^*$, this yields the following *upper bound* on suboptimality

$$f(x) - f^* \leq \frac{L}{s} d(x, X^*)^s. \tag{1}$$

Now, assume that $f$ satisfies the error bound (Sharp) on a set $K$ with parameters $(r, \mu)$. Combining (1) and (Sharp) this leads for every $x \in K$,

$$\frac{s\mu}{rL} \leq d(x, X^*)^{s-r}.$$

This means that necessarily $s \leq r$ by taking $x \to X^*$. Moreover if $s < r$, this last inequality can only be valid on a bounded set, i.e. either smoothness or error bound or both are valid only on a bounded set. In the following, we write

$$\kappa \triangleq L^{\frac{2}{s}} / \mu^{\frac{2}{r}} \qquad \text{and} \qquad \tau \triangleq 1 - \frac{s}{r} \tag{2}$$

respectively a generalized condition number for the function $f$ and a condition number based on the ratio of powers in inequalities (Smooth) and (Sharp). If $r = s = 2$, $\kappa$ matches the classical condition number of the function.

## 2 Scheduled restarts for smooth convex problems

In this section, we seek to solve (P) assuming that the function $f$ is smooth, i.e. satisfies (Smooth) with $s = 2$ and $L > 0$. Without further assumptions on $f$, an optimal algorithm to solve the smooth convex optimization problem (P) is Nesterov's accelerated gradient method [Nesterov, 1983]. Given an initial point $x_0$, this algorithm outputs, after $t$ iterations, a point $x = \mathcal{A}(x_0, t)$ such that

$$f(x) - f^* \leq \frac{cL}{t^2} d(x_0, X^*)^2, \tag{3}$$

where $c > 0$ denotes a universal constant (whose value will be allowed to vary in what follows, with $c = 4$ here). We assume without loss of generality that $f(x) \leq f(x_0)$. More details about Nesterov's algorithm are given in Supplementary Material.

In what follows, we will also assume that $f$ satisfies (Sharp) with parameters $(r, \mu)$ on a set $K \supseteq X^*$, which means

$$\frac{\mu}{r} d(x, X^*)^r \leq f(x) - f^*, \quad \text{for every } x \in K. \tag{Sharp}$$

As mentioned before if $r > s = 2$, this property is necessarily local, i.e. $K$ is bounded. We assume then that given a starting point $x_0 \in \mathbb{R}^n$, sharpness is satisfied on the sublevel set $\{x \mid f(x) \leq f(x_0)\}$. Remark that if this property is valid on an open set $K \supset X^*$, it will also be valid on any compact set $K' \supset K$ with the same exponent $r$ but a potentially lower constant $\mu$. The scheduled restart schemes we present here rely on a global sharpness hypothesis on the sublevel set defined by the initial point and are not adaptive to constant $\mu$ on smaller sublevel sets. On the other hand, restarts on criterion that we present in Section 4, assuming that $f^*$ is known, adapt to the value of $\mu$. We now describe a restart scheme exploiting this extra regularity assumption to improve the computational complexity of solving problem (P) using accelerated methods.

## 2.1 Scheduled restarts

Here, we schedule the number of iterations $t_k$ made by Nesterov's algorithm between restarts, with $t_k$ the number of (inner) iterations at the $k^{\text{th}}$ algorithm run (outer iteration). Our scheme is described in Algorithm 1 below.

---

**Algorithm 1** Scheduled restarts for smooth convex minimization

---

**Inputs :** $x_0 \in \mathbb{R}^n$ and a sequence $t_k$ for $k = 1, \ldots, R$.
**for** $k = 1, \ldots, R$ **do**
$$x_k := \mathcal{A}(x_{k-1}, t_k)$$
**end for**
**Output :** $\hat{x} := x_R$

---

The analysis of this scheme and the following ones relies on two steps. We first choose schedules that ensure linear convergence in the iterates $x_k$ at a given rate. We then adjust this linear rate to minimize the complexity in terms of the total number of iterations.

We begin with a technical lemma which assumes linear convergence holds, and connects the growth of $t_k$, the precision reached and the total number of inner iterations $N$.

**Lemma 2.1.** *Let $x_k$ be a sequence whose $k^{th}$ iterate is generated from the previous one by an algorithm that runs $t_k$ iterations and write $N = \sum_{k=1}^{R} t_k$ the total number of iterations to output a point $x_R$. Suppose setting $t_k = Ce^{\alpha k}$, $k = 1, \ldots, R$ for some $C > 0$ and $\alpha \geq 0$ ensures that outer iterations satisfy*

$$f(x_k) - f^* \leq \nu e^{-\gamma k}, \tag{4}$$

*for all $k \geq 0$ with $\nu \geq 0$ and $\gamma \geq 0$. Then precision at the output is given by,*

$$f(x_R) - f^* \leq \nu \exp(-\gamma N / C), \quad \text{when } \alpha = 0,$$

*and*

$$f(x_R) - f^* \leq \frac{\nu}{(\alpha e^{-\alpha} C^{-1} N + 1)^{\frac{\gamma}{\alpha}}}, \quad \text{when } \alpha > 0.$$

*Proof.* When $\alpha = 0$, $N = RC$, and inserting this in (4) at the last point $x_R$ yields the desired result. On the other hand, when $\alpha > 0$, we have $N = \sum_{k=1}^{R} t_k = Ce^{\alpha} \frac{e^{\alpha R} - 1}{e^{\alpha} - 1}$, which gives $R = \log\left(\frac{e^{\alpha} - 1}{e^{\alpha} C} N + 1\right) / \alpha$. Inserting this in (4) at the last point, we get

$$f(x_R) - f^* \leq \nu \exp\left(-\frac{\gamma}{\alpha} \log\left(\frac{e^{\alpha} - 1}{e^{\alpha} C} N + 1\right)\right) \leq \frac{\nu}{(\alpha e^{-\alpha} C^{-1} N + 1)^{\frac{\gamma}{\alpha}}},$$

where we used $e^x - 1 \geq x$. This yields the second part of the result. ∎

The last approximation in the case $\alpha > 0$ simplifies the analysis that follows without significantly affecting the bounds. We also show in Supplementary Material that using $\tilde{t}_k = \lceil t_k \rceil$ does not significantly affect the bounds above. Remark that convergence bounds are generally linear or polynomial such that we can extract a subsequence that converges linearly. Therefore our approach does not restrict the analysis of our scheme. It simplifies it and can be used for other algorithms like the gradient descent as detailed in Supplementary Material.

We now analyze restart schedules $t_k$ that ensure linear convergence. Our choice of $t_k$ will heavily depend on the ratio between $r$ and $s$ (with $s = 2$ for smooth functions here), incorporated in the parameter $\tau = 1 - s/r$ defined in (2). Below, we show that if $\tau = 0$, a constant schedule is sufficient to ensure linear convergence. When $\tau > 0$, we need a geometrically increasing number of iterations for each cycle.

**Proposition 2.2.** *Let $f$ be a smooth convex function satisfying* (Smooth) *with parameters* $(2, L)$ *and* (Sharp) *with parameters* $(r, \mu)$ *on a set $K$. Assume that we are given $x_0 \in \mathbb{R}^n$ such that $\{x| f(x) \leq f(x_0)\} \subset K$. Run Algorithm 1 from $x_0$ with iteration schedule $t_k = C_{\kappa,\tau}^* e^{\tau k}$, for $k = 1, \ldots, R$, where*

$$C_{\kappa,\tau}^* \triangleq e^{1-\tau} (c\kappa)^{\frac{1}{2}} (f(x_0) - f^*)^{-\frac{\tau}{2}}, \tag{5}$$

with $\kappa$ and $\tau$ defined in (2) and $c = 4e^{2/e}$ here. The precision reached at the last point $\hat{x}$ is given by,

$$f(\hat{x}) - f^* \leq \exp\left(-2e^{-1}(c\kappa)^{-\frac{1}{2}}N\right)(f(x_0) - f^*) = O\left(\exp(-\kappa^{-\frac{1}{2}}N)\right), \quad \text{when } \tau = 0, \quad (6)$$

while,

$$f(\hat{x}) - f^* \leq \frac{f(x_0) - f^*}{\left(\tau e^{-1}(f(x_0) - f^*)^{\frac{\tau}{2}}(c\kappa)^{-\frac{1}{2}}N + 1\right)^{\frac{2}{\tau}}} = O\left(N^{-\frac{2}{\tau}}\right), \quad \text{when } \tau > 0, \quad (7)$$

where $N = \sum_{k=1}^{R} t_k$ is the total number of iterations.

*Proof.* Our strategy is to choose $t_k$ such that the objective is linearly decreasing, i.e.

$$f(x_k) - f^* \leq e^{-\gamma k}(f(x_0) - f^*), \quad (8)$$

for some $\gamma \geq 0$ depending on the choice of $t_k$. This directly holds for $k = 0$ and any $\gamma \geq 0$. Combining (Sharp) with the complexity bound in (3), we get

$$f(x_k) - f^* \leq \frac{c\kappa}{t_k^2}(f(x_{k-1}) - f^*)^{\frac{2}{\tau}},$$

where $c = 4e^{2/e}$ using that $r^{2/r} \leq e^{2/e}$. Assuming recursively that (8) is satisfied at iteration $k-1$ for a given $\gamma$, we have

$$f(x_k) - f^* \leq \frac{c\kappa e^{-\gamma \frac{2}{\tau}(k-1)}}{t_k^2}(f(x_0) - f^*)^{\frac{2}{\tau}},$$

and to ensure (8) at iteration $k$, we impose

$$\frac{c\kappa e^{-\gamma \frac{2}{\tau}(k-1)}}{t_k^2}(f(x_0) - f^*)^{\frac{2}{\tau}} \leq e^{-\gamma k}(f(x_0) - f^*).$$

Rearranging terms in this last inequality, using $\tau$ defined in (2), we get

$$t_k \geq e^{\frac{\gamma(1-\tau)}{2}}(c\kappa)^{\frac{1}{2}}(f(x_0) - f^*)^{-\frac{\tau}{2}}e^{\frac{\tau\gamma}{2}k}. \quad (9)$$

For a given $\gamma \geq 0$, we can set $t_k = Ce^{\alpha k}$ where

$$C = e^{\frac{\gamma(1-\tau)}{2}}(c\kappa)^{\frac{1}{2}}(f(x_0) - f^*)^{-\frac{\tau}{2}} \quad \text{and} \quad \alpha = \tau\gamma/2, \quad (10)$$

and Lemma 2.1 then yields,

$$f(\hat{x}) - f^* \leq \exp\left(-\gamma e^{-\frac{\gamma}{2}}(c\kappa)^{-\frac{1}{2}}N\right)(f(x_0) - f^*),$$

when $\tau = 0$, while

$$f(\hat{x}) - f^* \leq \frac{(f(x_0) - f^*)}{\left(\frac{\tau}{2}\gamma e^{-\frac{\gamma}{2}}(c\kappa)^{-\frac{1}{2}}(f(x_0) - f^*)^{\frac{\tau}{2}}N + 1\right)^{\frac{2}{\tau}}},$$

when $\tau > 0$. These bounds are minimal for $\gamma = 2$, which yields the desired result. ∎

When $\tau = 0$, bound (6) matches the classical complexity bound for smooth strongly convex functions [Nesterov, 2013b]. When $\tau > 0$ on the other hand, bound (7) highlights a *much faster convergence rate than accelerated gradient methods*. The sharper the function (i.e. the smaller $r$), the faster the convergence. This matches the lower bounds for optimizing smooth and sharp functions functions [Arjevani and Shamir, 2016; Nemirovskii and Nesterov, 1985, Page 6] up to constant factors. Also, setting $t_k = C^*_{\kappa,\tau}e^{\tau k}$ yields continuous bounds on precision, i.e. when $\tau \to 0$, bound (7) converges to bound (6), which also shows that for $\tau$ near zero, constant restart schemes are almost optimal.

## 2.2 Adaptive scheduled restart

The previous restart schedules depend on the sharpness parameters $(r, \mu)$ in (Sharp). In general of course, these values are neither observed nor known a priori. Making our restart scheme adaptive is thus crucial to its practical performance. Fortunately, we show below that a simple logarithmic grid search strategy on these parameters is enough to guarantee nearly optimal performance.

We run several schemes with a fixed number of inner iterations $N$ to perform a log-scale grid search on $\tau$ and $\kappa$. We define these schemes as follows.

$$\begin{cases} \mathcal{S}_{i,0} : \text{Algorithm 1 with } t_k = C_i, \\ \mathcal{S}_{i,j} : \text{Algorithm 1 with } t_k = C_i e^{\tau_j k}, \end{cases} \tag{11}$$

where $C_i = 2^i$ and $\tau_j = 2^{-j}$. We stop these schemes when the total number of inner algorithm iterations has exceed $N$, i.e. at the smallest $R$ such that $\sum_{k=1}^{R} t_k \geq N$. The size of the grid search in $C_i$ is naturally bounded as we cannot restart the algorithm after more than $N$ total inner iterations, so $i \in [1, \ldots, \lfloor \log_2 N \rfloor]$. We will also show that when $\tau$ is smaller than $1/N$, a constant schedule performs as well as the optimal geometrically increasing schedule, which crucially means we can also choose $j \in [1, \ldots, \lceil \log_2 N \rceil]$ and limits the cost of grid search. The following result details the convergence of this method, its notations are the same as in Proposition 2.2 and its technical proof can be found in Supplementary Material.

**Proposition 2.3.** *Let $f$ be a smooth convex function satisfying (Smooth) with parameters $(2, L)$ and (Sharp) with parameters $(r, \mu)$ on a set $K$. Assume that we are given $x_0 \in \mathbb{R}^n$ such that $\{x \mid f(x) \leq f(x_0)\} \subset K$ and denote $N$ a given number of iterations. Run schemes $\mathcal{S}_{i,j}$ defined in (11) to solve (P) for $i \in [1, \ldots, \lfloor \log_2 N \rfloor]$ and $j \in [0, \ldots, \lceil \log_2 N \rceil]$, stopping each time after $N$ total inner algorithm iterations i.e. for $R$ such that $\sum_{k=1}^{R} t_k \geq N$.*

*Assume $N$ is large enough, so $N \geq 2C_{\kappa,\tau}^*$, and if $\frac{1}{N} > \tau > 0$, $C_{\kappa,\tau}^* > 1$.*

*If $\tau = 0$, there exists $i \in [1, \ldots, \lfloor \log_2 N \rfloor]$ such that scheme $\mathcal{S}_{i,0}$ achieves a precision given by*

$$f(\hat{x}) - f^* \leq \exp\left(-e^{-1}(c\kappa)^{-\frac{1}{2}} N\right) (f(x_0) - f^*).$$

*If $\tau > 0$, there exist $i \in [1, \ldots, \lfloor \log_2 N \rfloor]$ and $j \in [1, \ldots, \lceil \log_2 N \rceil]$ such that scheme $\mathcal{S}_{i,j}$ achieves a precision given by*

$$f(\hat{x}) - f^* \leq \frac{f(x_0) - f^*}{\left(\tau e^{-1}(c\kappa)^{-\frac{1}{2}}(f(x_0) - f^*)^{\frac{\tau}{2}}(N-1)/4 + 1\right)^{\frac{2}{\tau}}}.$$

*Overall, running the logarithmic grid search has a complexity $(\log_2 N)^2$ times higher than running $N$ iterations using the optimal (oracle) scheme.*

As showed in Supplementary Material, scheduled restart schemes are theoretically efficient only if the algorithm itself makes a sufficient number of iterations to decrease the objective value. Therefore we need $N$ large enough to ensure the efficiency of the adaptive method. If $\tau = 0$, we naturally have $C_{\kappa,0}^* \geq 1$, therefore if $\frac{1}{N} > \tau > 0$ and $N$ is large, assuming $C_{\kappa,\tau}^* \approx C_{\kappa,0}^*$, we get $C_{\kappa,\tau}^* \geq 1$. This adaptive bound is similar to the one of Nesterov [2013b] to optimize smooth strongly convex functions in the sense that we lose approximately a log factor of the condition number of the function. However our assumptions are weaker and we are able to tackle all regimes of the sharpness property, i.e. any exponent $r \in [2, +\infty]$, not just the strongly convex case.

In the supplementary material we also analyze the simple gradient descent method under the sharpness (Sharp) assumption. It shows that simple gradient descent achieves a $O(\epsilon^{-\tau})$ complexity for a given accuracy $\epsilon$. Therefore restarting accelerated gradient methods reduces complexity to $O(\epsilon^{-\tau/2})$ compared to simple gradient descent. This result is similar to the acceleration of gradient descent. We extend now this restart scheme to solve non-smooth or Hölder smooth convex optimization problem under the sharpness assumption.

## 3 Universal scheduled restarts for convex problems

In this section, we use the framework introduced by Nesterov [2015] to describe smoothness of a convex function $f$, namely, we assume that there exist $s \in [1, 2]$ and $L > 0$ on a set $J \subset \mathbb{R}^n$, i.e.

$$\|\nabla f(x) - \nabla f(y)\| \leq L\|x - y\|^{s-1}, \quad \text{for every } x, y \in J.$$

Without further assumptions on $f$, the optimal rate of convergence for this class of functions is bounded as $O(1/N^\rho)$, where $N$ is the total number of iterations and

$$\rho = 3s/2 - 1, \tag{12}$$

which gives $\rho = 2$ for smooth functions and $\rho = 1/2$ for non-smooth functions. The universal fast gradient method [Nesterov, 2015] achieves this rate by requiring only a target accuracy $\epsilon$ and a starting point $x_0$. It outputs after $t$ iterations a point $x \triangleq \mathcal{U}(x_0, \epsilon, t)$, such that

$$f(x) - f^* \le \frac{\epsilon}{2} + \frac{cL^{\frac{2}{s}}d(x_0, X^*)^2}{\epsilon^{\frac{2}{s}}t^{\frac{2\rho}{s}}}\frac{\epsilon}{2}, \tag{13}$$

where $c$ is a constant ($c = 2^{\frac{4s-2}{s}}$). More details about the universal fast gradient method are given in Supplementary Material.

We will again assume that $f$ is sharp with parameters $(r, \mu)$ on a set $K \supseteq X^*$, i.e.

$$\frac{\mu}{r}d(x, X^*)^r \le f(x) - f^*, \quad \text{for every } x \in K. \tag{Sharp}$$

As mentioned in Section 1.2, if $r > s$, smoothness or sharpness are local properties, i.e. either $J$ or $K$ or both are bounded, our analysis is therefore local. In the following we assume for simplicity, given an initial point $x_0$, that smoothness and sharpness are satisfied simultaneously on the sublevel set $\{x \mid f(x) \le f(x_0)\}$. The key difference with the smooth case described in the previous section is that here we schedule *both* the target accuracy $\epsilon_k$ used by the algorithm *and* the number of iterations $t_k$ made at the $k^{\text{th}}$ run of the algorithm. Our scheme is described in Algorithm 2.

---

**Algorithm 2** Universal scheduled restarts for convex minimization

**Inputs :** $x_0 \in \mathbb{R}^n$, $\epsilon_0 \ge f(x_0) - f^*$, $\gamma \ge 0$ and a sequence $t_k$ for $k = 1, \ldots, R$.
**for** $k = 1, \ldots, R$ **do**
$$\epsilon_k := e^{-\gamma}\epsilon_{k-1}, \qquad x_k := \mathcal{U}(x_{k-1}, \epsilon_k, t_k)$$
**end for**
**Output :** $\hat{x} := x_R$

---

Our strategy is to choose a sequence $t_k$ that ensures

$$f(x_k) - f^* \le \epsilon_k,$$

for the geometrically decreasing sequence $\epsilon_k$. The overall complexity of our method will then depend on the growth of $t_k$ as described in Lemma 2.1. The proof is similar to the smooth case and can be found in Supplementary Material.

**Proposition 3.1.** *Let $f$ be a convex function satisfying* (Smooth) *with parameters $(s, L)$ on a set $J$ and* (Sharp) *with parameters $(r, \mu)$ on a set $K$. Given $x_0 \in \mathbb{R}^n$ assume that $\{x \mid f(x) \le f(x_0)\} \subset J \cap K$. Run Algorithm 2 from $x_0$ for a given $\epsilon_0 \ge f(x_0) - f^*$ with*

$$\gamma = \rho, \qquad t_k = C^*_{\kappa, \tau, \rho}e^{\tau k}, \quad \text{where} \quad C^*_{\kappa, \tau, \rho} \triangleq e^{1-\tau}(c\kappa)^{\frac{s}{2\rho}}\epsilon_0^{-\frac{\tau}{\rho}}$$

*where $\rho$ is defined in* (12)*, $\kappa$ and $\tau$ are defined in* (2) *and $c = 8e^{2/e}$ here. The precision reached at the last point $\hat{x}$ is given by,*

$$f(\hat{x}) - f^* \le \exp\left(-\rho e^{-1}(c\kappa)^{-\frac{s}{2\rho}}N\right)\epsilon_0 = O\left(\exp(-\kappa^{-\frac{s}{2\rho}}N)\right), \quad \text{when } \tau = 0,$$

*while,*

$$f(\hat{x}) - f^* \le \frac{\epsilon_0}{\left(\tau e^{-1}(c\kappa)^{-\frac{s}{2\rho}}\epsilon_0^{\frac{\tau}{\rho}}N + 1\right)^{-\frac{\rho}{\tau}}} = O\left(\kappa^{\frac{s}{2\tau}}N^{-\frac{\rho}{\tau}}\right), \quad \text{when } \tau > 0,$$

*where $N = \sum_{k=1}^{R} t_k$ is total number of iterations.*

This bound matches the lower bounds for optimizing smooth and sharp functions [Nemirovskii and Nesterov, 1985, Page 6] up to constant factors. Notice that, compared to Nemirovskii and Nesterov [1985], we can tackle non-smooth convex optimization by using the universal fast gradient algorithm of Nesterov [2015]. The rate of convergence in Proposition 3.1 is controlled by the ratio between $\tau$ and $\rho$. If these are unknown, a log-scale grid search won't be able to reach the optimal rate, even if $\rho$ is known since we will miss the optimal rate by a constant factor. If both are known, in the case of non-smooth strongly convex functions for example, a grid-search on $C$ recovers nearly the optimal bound. Now we will see that if $f^*$ is known, restart produces adaptive optimal rates.

# 4   Restart with termination criterion

Here, we assume that we know the optimum $f^*$ of (P), or have an exact termination criterion. This is the case for example in zero-sum matrix games problems or non-degenerate least-squares without regularization. We assume again that $f$ satisfies (Smooth) with parameters $(s, L)$ on a set $J$ and (Sharp) with parameters $(r, \mu)$ on a set $K$. Given an initial point $x_0$ we assume that smoothness and sharpness are satisfied simultaneously on the sublevel set $\{x \mid f(x) \leq f(x_0)\}$. We use again the universal gradient method $\mathcal{U}$. Here however, we can stop the algorithm when it reaches the target accuracy as we know the optimum $f^*$, i.e. we stop after $t_\epsilon$ inner iterations such that $x = \mathcal{U}(x_0, \epsilon, t_\epsilon)$ satisfies $f(x) - f^* \leq \epsilon$, and write $x \triangleq \mathcal{C}(x_0, \epsilon)$ the output of this method.

Here we simply restart this method and decrease the target accuracy by a constant factor after each restart. Our scheme is described in Algorithm 3.

---

**Algorithm 3** Restart on criterion

---

**Inputs :** $x_0 \in \mathbb{R}^n, f^*, \gamma \geq 0, \epsilon_0 = f(x_0) - f^*$
**for** $k = 1, \ldots, R$ **do**

$$\epsilon_k := e^{-\gamma}\epsilon_{k-1}, \qquad x_k := \mathcal{C}(x_{k-1}, \epsilon_k)$$

**end for**
**Output :** $\hat{x} := x_R$

---

The following result describes the convergence of this method. It relies on the idea that it cannot do more iterations than the best scheduled restart to achieve the target accuracy at each iteration. Its proof can be found in Supplementary Material.

**Proposition 4.1.** *Let $f$ be a convex function satisfying* (Smooth) *with parameters $(s, L)$ on a set $J$ and* (Sharp) *with parameters $(r, \mu)$ on a set $K$. Given $x_0 \in \mathbb{R}^n$ assume that $\{x, f(x) \leq f(x_0)\} \subset J \cap K$. Run Algorithm 3 from $x_0$ with parameter $\gamma = \rho$. The precision reached at the last point $\hat{x}$ is given by,*

$$f(\hat{x}) - f^* \leq \exp\left(-\rho e^{-1}(c\kappa)^{-\frac{s}{2\rho}}N\right)(f(x_0) - f^*) = O\left(\exp(-\kappa^{-\frac{s}{2\rho}}N)\right), \quad \text{when } \tau = 0,$$

*while,*

$$f(\hat{x}) - f^* \leq \frac{f(x_0) - f^*}{\left(\tau e^{-1}(c\kappa)^{-\frac{s}{2\rho}}(f(x_0) - f^*)^{\frac{\tau}{\rho}}N + 1\right)^{\frac{\rho}{\tau}}} = O\left(\kappa^{\frac{s}{2\tau}}N^{-\frac{\rho}{\tau}}\right), \quad \text{when } \tau > 0,$$

*where $N$ is the total number of iterations, $\rho$ is defined in (12), $\kappa$ and $\tau$ are defined in (2) and $c = 8e^{2/e}$ here.*

Therefore if $f^*$ is known, this method is adaptive, contrary to the general case in Proposition 3.1. It can even adapt to the local values of $L$ or $\mu$ as we use a criterion instead of a preset schedule. Here, stopping using $f(x_k) - f^*$ implicitly yields optimal choices of $C$ and $\tau$. A closer look at the proof shows that the dependency in $\gamma$ of this restart scheme is a factor $h(\gamma) = \gamma e^{-\gamma/\rho}$ of the number of iterations. Taking $\gamma = 1$, leads then to a suboptimal constant factor of at most $h(\rho)/h(1) \leq e/2 \approx 1.3$ for $\rho \in [1/2, 2]$, so running this scheme with $\gamma = 1$ makes it parameter-free while getting nearly optimal bounds.

# 5   Numerical Results

We illustrate our results by testing our adaptive restart methods, denoted *Adap* and *Crit*, introduced respectively in Sections 2.2 and 4 on several problems and compare them against simple gradient descent (*Grad*), accelerated gradient methods (*Acc*), and the restart heuristic enforcing monotonicity (*Mono* in [O'Donoghue and Candes, 2015]). For *Adap* we plot the convergence of the best method found by grid search to compare with the restart heuristic. This implicitly assumes that the grid search is run in parallel with enough servers. For *Crit* we use the optimal $f^*$ found by another solver. This gives an overview of its performance in order to potentially approximate it along the iterations

in a future work as done with Polyak steps [Polyak, 1987]. All restart schemes were done using the accelerated gradient with backtracking line search detailed in the Supplementary Material, with large dots representing restart iterations.

The results focused on unconstrained problems but our approach can directly be extended to composite problems by using the proximal variant of the gradient, accelerated gradient and universal fast gradient methods [Nesterov, 2015] as detailed in the Supplementary Material. This includes constrained optimization as a particular case by adding the indicator function of the constraint set to the objective (as in the SVM example below).

In Figure 1, we solve classification problems with various losses on the UCI *Sonar* data set [Asuncion and Newman, 2007]. For least square loss on sonar data set, we observe much faster convergence of the restart schemes compared to the accelerated method. These results were already observed by O'Donoghue and Candes [2015]. For logistic loss, we observe that restart does not provide much improvement. The backtracking line search on the Lipschitz constant may be sufficient to capture the geometry of the problem. For hinge loss, we regularized by a squared norm and optimize the dual, which means solving a quadratic problem with box constraints. We observe here that the scheduled restart scheme convergences much faster, while restart heuristics may be activated too late. We observe similar results for the LASSO problem. In general *Crit* ensures the theoretical accelerated rate but *Adap* exhibits more consistent behavior. This highlights the benefits of a sharpness assumption for these last two problems. Precisely quantifying sharpness from data/problem structure is a key open problem.

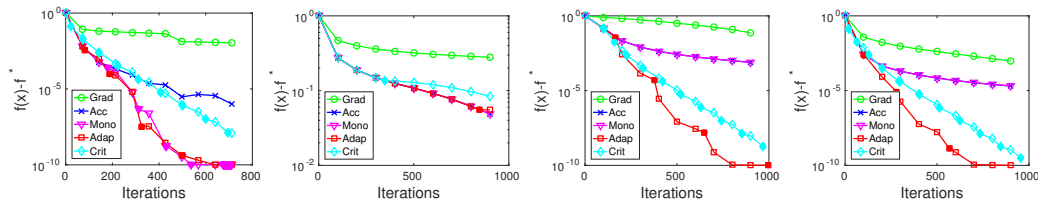

Figure 1: From left to right: least square loss, logistic loss, dual SVM problem and LASSO. We use adaptive restarts (Adap), gradient descent (Grad), accelerated gradient (Acc) and restart heuristic enforcing monotonicity (Mono). Large dots represent the restart iterations. Regularization parameters for dual SVM and LASSO were set to one.

## Acknowledgments

The authors would like to acknowledge support from the chaire *Économie des nouvelles données* with the *data science* joint research initiative with the *fonds AXA pour la recherche*, a gift from Société Générale Cross Asset Quantitative Research and an AMX fellowship. The authors are affiliated to PSL Research University, Paris, France.

## Footnotes

[1]We assume the problem feasible, i.e. $X^* \neq \emptyset$.

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
