[Supplementary Material]

# Sharpness, Restart and Acceleration
# Supplementary Material

## Overview

This Supplementary Material is organized as follows. An analysis of simple gradient descent under the sharpness assumption is provided in Section 6. Restart schemes are extended to composite problems and non-Euclidean settings in Section 7. In Section 8 we precise the convergence results for integer schedules. Finally section 9 is dedicated to a complete presentation of the algorithms that we restart and missing proofs of convergence rates can be found in Section 10.

## 6 Comparison to gradient descent

Given only the smoothness hypothesis (Smooth) with parameters $(2, L)$, the gradient descent algorithm, recalled in Appendix 9.3, starts from a point $x_0$ and outputs iterates $x_t = \mathcal{G}(x_0, t)$ such that

$$f(x_t) - f^* \leq \frac{L}{t} d(x_0, X^*)^2,$$

While accelerated methods use the last two iterates to compute the next one, simple gradient descent algorithms use only the last iterate, so the algorithm can be seen as (implicitly) restarting at each iteration. Its convergence can therefore be written for $k \geq 1$,

$$f(x_{k+t}) - f^* \leq \frac{L}{t} d(x_k, X^*)^2. \tag{14}$$

and we analyze it in light of the restart interpretation using the sharpness assumption in the following proposition.

**Proposition 6.1.** *Let $f$ be a smooth convex function satisfying (Smooth) with parameters $(2, L)$ and (Sharp) with parameters $(r, \mu)$ on a set $K$. Assume that we are given $x_0 \in \mathbb{R}^n$ such that $\{x \mid f(x) \leq f(x_0)\} \subset K$. Denote $x_t = \mathcal{G}(x_0, t)$ the iterate sequence generated by the gradient descent algorithm started at $x_0$ to solve (P). Define*

$$t_k = e^{1-\tau} c\kappa (f(x_0) - f^*)^\tau e^{\tau k},$$

*with $\kappa$ and $\tau$ defined in (2) and $c = e^{2/e}$ here. The precision reached after $N = \sum_{k=1}^n t_k$ iterations is given by,*

$$f(x_N) - f^* \leq \exp\left(-e^{-1}(c\kappa)^{-1}N\right)(f(x_0) - f^*) = O\left(\exp(-\kappa^{-1}N)\right), \quad \text{when } \tau = 0,$$

*while,*

$$f(x_N) - f^* \leq \frac{f(x_0) - f^*}{(\tau e^{-1}(c\kappa)^{-1}(f(x_0) - f^*)^\tau N + 1)^{\frac{1}{\tau}}} = O\left(N^{-\frac{1}{\tau}}\right), \quad \text{when } \tau > 0.$$

*Proof.* For a given $\gamma \geq 0$, we construct a subsequence $x_{\phi(k)}$ of $x_t$ such that

$$f(x_{\phi(k)}) - f^* \leq e^{-\gamma k}(f(x_0) - f^*). \tag{15}$$

Define $x_{\phi(0)} = x_0$. Assume that (15) is true at iteration $k-1$, then combining complexity bound (14) and (Sharp), for any $t \geq 1$,

$$
\begin{aligned}
f(x_{\phi(k-1)+t}) - f^* &\leq \frac{c\kappa}{t}(f(x_{\phi(k-1)}) - f^*)^{\frac{2}{r}} \\
&\leq \frac{c\kappa}{t}e^{-\gamma\frac{2}{r}(k-1)}(f(x_0) - f^*)^{\frac{2}{r}}.
\end{aligned}
$$

where $c = e^{2/e}$, using that $r^{2/r} \leq e^{2/e}$. Taking $t_k = e^{\gamma(1-\tau)}c\kappa(f(x_0) - f^*)^{-\tau}e^{\gamma\tau k}$ and $\phi(k) = \phi(k-1) + t_k$, (15) holds at iteration $k$. Using Lemma 2.1, we obtain at iteration $N = \phi(n) = \sum_{k=1}^{n} t_k$,

$$
f(x_N) - f^* \leq \exp\left(-\gamma e^{-\gamma}(c\kappa)^{-1}N\right)(f(x_0) - f^*), \quad \text{if } \tau = 0,
$$

and

$$
f(x_N) - f^* \leq \frac{f(x_0) - f^*}{(\tau\gamma e^{-\gamma}(c\kappa)^{-1}(f(x_0) - f^*)^{\tau}N + 1)^{\frac{1}{\tau}}}, \quad \text{if } \tau > 0.
$$

These bounds are minimal for $\gamma = 1$ and the results follow. ∎

We observe that restarting accelerated gradient methods reduces complexity from $O(1/\epsilon^{\tau})$ to $O(1/\epsilon^{\tau/2})$ compared to simple gradient descent. More general results on the convergence of (sub)gradient descent algorithms under a Łojasiewicz inequality assumption were developed by Bolte et al. [2015].

## 7 Composite problems & Bregman divergences

The restart schemes detailed so far focused on unconstrained problems in an Euclidean setting. Here, we extend them to more general convex optimization problems of the form

$$
\text{minimize } f(x) \triangleq \phi(x) + g(x), \tag{Composite}
$$

where $\phi$ is a convex function whose smoothness is described by parameters $(L, s)$, such that

$$
\|\nabla\phi(x) - \nabla\phi(y)\|_* \leq L\|x - y\|^{s-1} \quad \text{for every } x, y \in J, \tag{Generic Smooth}
$$

for a given norm $\|\cdot\|$ where $\|\cdot\|_*$ is its dual norm, and $g$ is a simple convex function (the meaning of simple will be clarified later).

To exploit the smoothness of $\phi$ with respect to a generic norm, we assume that we have access to a prox function $h$ with $\mathbf{dom}(f) \subset \mathbf{dom}(h)$, strongly convex with respect to the norm $\|\cdot\|$ with convexity parameter equal to one, which means

$$
h(y) \geq h(x) + \nabla h(x)^T(y - x) + \frac{1}{2}\|x - y\|^2, \quad \text{for any } x, y \in \mathbf{dom}(h).
$$

We define the Bregman divergence associated to $h$ as

$$
D_h(y, x) = h(y) - h(x) - \nabla h(x)^T(y - x), \quad \text{for } x, y \in \mathbf{dom}(h),
$$

so that $D_h(y, x) \geq \frac{1}{2}\|x - y\|^2$. For $h(x) = \frac{1}{2}\|x\|_2^2$, we get $D_h(y, x) = \frac{1}{2}\|x - y\|_2^2$ and recover the Euclidean setting. Given the problem geometry, appropriate choices of prox functions and associated Bregman divergences can lead to significant performance gains in high dimensional settings.

We now formally state the assumption that $g$ is simple. Given $x, y \in \mathbf{dom}(f)$ and $\lambda \geq 0$ we assume that

$$
\min_z \left\{y^T z + g(z) + \lambda D_h(z, x)\right\}
$$

can be solved either in a closed form or by some fast computational procedure. Examples of such settings include sparse optimization problems, such as the LASSO, where $\phi(x) = \|Ax - b\|_2^2$, with $A \in \mathbb{R}^{m \times n}$, $b \in \mathbb{R}^m$, $g(x) = \lambda\|x\|_1$, with $\lambda \geq 0$ and $h(x) = \frac{1}{2}\|x\|_2^2$. This setting also includes constrained optimization problems, where $g$ is the indicator function of a closed convex set. We see in numerical experiments that restart schemes applied in these two settings lead to significant performance improvements.

To apply our analysis of restart schemes we need two things: an accelerated algorithm that tackles such setting and an appropriate notion of sharpness. We first introduce the notion of relative error bound.

**Definition 7.1.** *A convex function $f$ is called relatively sharp with respect to a strongly convex function $h$ on a set $K \subset \mathbf{dom}(f)$ iff there exist $r \geq 1$, $\mu > 0$ such that*

$$\frac{\mu}{r} D_h(x, X^*)^{\frac{r}{2}} \leq f(x) - f^* \quad \text{for any } x \in K \qquad \text{(Relative Sharp)}$$

*where $D_h(x, X^*) = \min_{x^* \in X^*} D_h(x, x^*)$ and $D_h$ is the Bregman divergence associated to $h$.*

If $h = \frac{1}{2}\|x\|_2^2$ we recover the definition of sharpness in the Euclidean setting (with slightly modified constants). This assumption is as generic as our first one in (Sharp) as it is satisfied if $f$ and $h$ are subanalytic [Bierstone and Milman, 1988, Th. 6.4].

The universal fast gradient is then the candidate in this setting. Given a target accuracy $\epsilon$ and an initial point $x_0$, it outputs, after $t$ iterations, a point $x = \mathcal{U}(x_0, \epsilon, t)$ such that

$$f(x) - f^* \leq \frac{\epsilon}{2} + \frac{cL^{\frac{2}{s}} D_h(x_0, X^*)}{\epsilon^{\frac{2}{s}} t^{\frac{2\rho}{s}}} \frac{\epsilon}{2},$$

where $c = 16$ here. All our previous results can then directly be transposed to the setting here, as their proofs rely only on this convergence bound. We restate them in this setting below. First if $\phi$ is known to be smooth ($s = 2$) the universal fast gradient algorithm simplifies as the accelerated gradient algorithm (see Appendix 9.2) and the next Corollary generalizes Proposition 2.2.

**Corollary 7.2.** *Let $\phi, g, h$ defining the composite problem (Composite) described above with $\phi$ satisfying (Generic Smooth) with parameters $(2, L)$ and $f = \phi + g$ satisfies the (Relative Sharp) condition with respect to $h$ with parameters $(r, \mu)$ on a set $K$. Assume that we are given $x_0 \in \mathbb{R}^n$ such that $\{x \mid f(x) \leq f(x_0)\} \subset K$. Run Algorithm 1 from $x_0$ with iteration schedule $t_k = C^*_{\kappa,\tau} e^{\tau k}$, for $k = 1, \ldots, R$, where*

$$C^*_{\kappa,\tau} \triangleq e^{1-\tau}(c\kappa)^{\frac{1}{2}}(f(x_0) - f^*)^{-\frac{\tau}{2}},$$

*with $\kappa$ and $\tau$ defined in (2) and $c = 8e^{2/e}$. The precision reached at the last point $\hat{x}$ is given by,*

$$f(\hat{x}) - f^* \leq \exp\left(-2e^{-1}(c\kappa)^{-\frac{1}{2}} N\right)(f(x_0) - f^*) = O\left(\exp(-\kappa^{-\frac{1}{2}} N)\right), \quad \text{when } \tau = 0,$$

*while,*

$$f(\hat{x}) - f^* \leq \frac{f(x_0) - f^*}{\left(\tau e^{-1}(f(x_0) - f^*)^{\frac{\tau}{2}}(c\kappa)^{-\frac{1}{2}} N + 1\right)^{\frac{2}{\tau}}} = O\left(\kappa^{\frac{1}{\tau}} N^{-\frac{2}{\tau}}\right), \quad \text{when } \tau > 0,$$

*where $N = \sum_{k=1}^{R} t_k$ is the total number of iterations.*

For general convex functions, the following Corollary generalizes Proposition 3.1.

**Corollary 7.3.** *Let $\phi, g, h$ defining the composite problem (Composite) described above with $\phi$ satisfying (Generic Smooth) with parameters $(s, L)$ on a set $J$ and $f = \phi + g$ satisfying (Relative Sharp) with respect to $h$ with parameters $(r, \mu)$ on a set $K$. Given $x_0 \in \mathbb{R}^n$ assume that $\{x \mid f(x) \leq f(x_0)\} \subset Q \cap K$. Run Algorithm 2 from $x_0$ for given $\epsilon_0 \geq f(x_0) - f^*$,*

$$\gamma = \rho, \qquad t_k = C^*_{\kappa,\tau,\rho} e^{\tau k}, \quad \text{where} \quad C^*_{\kappa,\tau,\rho} \triangleq e^{1-\tau}(c\kappa)^{\frac{s}{2\rho}} \epsilon_0^{-\frac{\tau}{\rho}}$$

*where $\rho$ is defined in (12), $\kappa$ and $\tau$ are defined in (2) and $c = 16e^{2/e}$. The precision reached at the last point $\hat{x}$ is given by,*

$$f(\hat{x}) - f^* \leq \exp\left(-\rho e^{-1}(c\kappa)^{-\frac{s}{2\rho}} N\right) \epsilon_0 = O\left(\exp(-\kappa^{-\frac{s}{2\rho}} N)\right), \quad \text{when } \tau = 0,$$

*while,*

$$f(\hat{x}) - f^* \leq \frac{\epsilon_0}{\left(\tau e^{-1}(c\kappa)^{-\frac{s}{2\rho}} \epsilon_0^{\frac{\rho}{\rho}} N + 1\right)^{\frac{\rho}{\tau}}} = O\left(\kappa^{\frac{s}{2\tau}} N^{-\frac{\rho}{\tau}}\right), \quad \text{when } \tau > 0,$$

*where $N = \sum_{k=1}^{R} t_k$ is total number of iterations.*

The results regarding adaptive schemes and those with termination criterion generalize similarly under the relative sharpness assumption.

# 8 Rounding issues

We presented convergence bounds for real sequences of iterate counts $(t_k)_{k=1}^{\infty}$ but in practice these are integer sequences. The following Lemma details the convergence of our schemes for an approximate choice $\tilde{t}_k = \lceil t_k \rceil$.

**Lemma 8.1.** *Let $x_k$ be a sequence whose $k^{th}$ iterate is generated from previous one by an algorithm that needs $t_k$ iterations and denote $N = \sum_{k=1}^{R} t_k$ the total number of iterations to output a point $\hat{x} = x_R$. Suppose setting*

$$t_k = \lceil Ce^{\alpha k} \rceil, \quad k = 1, \ldots, R$$

*for some $C > 0$ and $\alpha \geq 0$ ensures that objective values $f(x_k)$ converge linearly, i.e.*

$$f(x_k) - f^* \leq \nu e^{-\gamma k}, \tag{16}$$

*for all $k \geq 0$ with $\nu \geq 0$ and $\gamma \geq 0$. Then precision at the output is given by,*

$$f(\hat{x}) - f^* \leq \nu \exp(-\gamma N/(C+1)), \quad \text{when } \alpha = 0,$$

*and*

$$f(\hat{x}) - f^* \leq \frac{\nu}{(\alpha e^{-\alpha} C^{-1} N' + 1)^{\frac{\gamma}{\alpha}}}, \quad \text{when } \alpha > 0,$$

*where $N' = N - \frac{\log\left((e^{\alpha}-1)e^{-\alpha}C^{-1}N+1\right)}{\alpha}$.*

*Proof.* At the $R^{\text{th}}$ point generated, $N = \sum_{k=1}^{R} t_k$. If $t_k = \lceil C \rceil$, define $\epsilon = \lceil C \rceil - C$ such that $0 \leq \epsilon < 1$. Then $N = R(C + \epsilon)$, injecting it in (16) at the $R^{\text{th}}$ point, we get

$$f(\hat{x}) - f^* \leq \nu e^{-\gamma \frac{N}{C+\epsilon}} \leq \nu e^{-\gamma \frac{N}{C+1}}.$$

Now, if $t_k = \lceil Ce^{\alpha k} \rceil$, define $\epsilon_k = \lceil Ce^{\alpha k} \rceil - Ce^{\alpha k}$, such that $0 \leq \epsilon_k < 1$. On one hand

$$N \geq \sum_{k=1}^{R} Ce^{\alpha k},$$

such that

$$R \leq \frac{\log\left((e^{\alpha} - 1)e^{-\alpha}C^{-1}N + 1\right)}{\alpha}.$$

On the other hand,

$$
\begin{aligned}
N = \sum_{k=1}^{R} t_k &= \frac{Ce^{\alpha}}{e^{\alpha} - 1}(e^{\alpha R} - 1) + \sum_{k=1}^{R} \epsilon_k \\
&\leq \frac{Ce^{\alpha}}{e^{\alpha} - 1}(e^{\alpha R} - 1) + R \\
&\leq \frac{Ce^{\alpha}}{e^{\alpha} - 1}(e^{\alpha R} - 1) + \frac{\log\left((e^{\alpha} - 1)e^{-\alpha}C^{-1}N + 1\right)}{\alpha},
\end{aligned}
$$

such that

$$R \geq \frac{\log\left(\alpha e^{-\alpha}C^{-1}N' + 1\right)}{\alpha}.$$

Injecting it in (16) at the $R^{\text{th}}$ point we get the result. ■

# 9 Algorithms & Complexity Bounds

We present here the classical algorithms for convex optimization that we restart. We present their general form to solve composite optimization problems of the form

$$\text{minimize} \quad f(x) = \phi(x) + g(x) \tag{Composite}$$

where $\phi, g$ are convex functions and $g$ is assumed simple. This setting is detailed in Section 7.

## 9.1 Universal fast gradient method

An optimal algorithm to solve the (Composite) problem is then the universal fast gradient method [Nesterov, 2015]. It is detailed in Algorithm 4. Given a target accuracy $\epsilon$, it starts at a point $x_0$ and outputs after $t$ iterations a point $x \triangleq \mathcal{U}(x_0, \epsilon, t)$, such that

$$f(x) - f^* \leq \frac{\epsilon}{2} + \frac{cL^{\frac{2}{s}} D_h(x_0, X^*)}{\epsilon^{\frac{2}{s}} t^{\frac{2\rho}{s}}} \frac{\epsilon}{2},$$

where $D_h(x; X^*) = \min_{x^* \in X^*} D_h(x; x^*)$ is the Bregman distance from $x$ to the set of minimizers, $c$ is a constant ($c = 2^{\frac{5s-2}{s}}$) and $\rho = \frac{3s-2}{2}$ is the optimal rate of convergence as presented in Section 3. In the Euclidean setting, $h = \frac{1}{2}\|x\|_2^2$, $D_h(y; x) = \frac{1}{2}\|x - y\|^2$, such that we get the bound given in (13).

The method does not need to know the smoothness parameters $(s, L)$, but the target accuracy $\epsilon$ is used to parametrize the algorithm. The universal fast gradient method requires an estimate $L_0$ of the smoothness parameter $L$ to start a line search on $L$. This line search is proven to increase the complexity of the algorithm by at most a constant factor plus a logarithmic term and ensures that the overall complexity does not depend on $L_0$ but on $L$. In our restart schemes we use a first estimate $L_0$ when running the algorithm for the first time and we use the last estimate found by the algorithm when restarting it.

Finally if the problem is feasible ($X^* \neq \emptyset$), the universal fast gradient method produces a convergent sequence of iterates. Therefore if the Łojasiewicz inequality is satisfied on a compact set $K$, it will be valid for all our iterates after perhaps reducing $\mu$.

---

**Algorithm 4** Universal fast gradient method

**Inputs :** $x_0$, $L_0$, $\epsilon$
**Initialize :** $y_0 := x_0$, $A_0 := 0$, $\hat{L} := L_0$
**for** $t = 0, \ldots, T$ **do**

$$z_t := \arg\min_z \sum_{i=1}^{t} a_i \nabla\phi(x_i)^T z + A_t g(z) + D_h(z; x_0)$$

   **repeat**
      Find $a \geq 0$, such that

$$a^2 = \frac{1}{\hat{L}}(A_t + a)$$

     Choose

$$
\begin{aligned}
\tau &:= \frac{a}{A_t + a} \\
x &:= \tau z_t + (1 - \tau)y_t \\
\hat{x} &:= \arg\min_z a\nabla\phi(x)^T z + a\psi(z) + D_h(z; z_t) \\
y &:= \tau\hat{x} + (1 - \tau)y_t
\end{aligned}
$$

   **if** $\phi(y) \geq \phi(x) + \langle\nabla\phi(x), y - x\rangle + \frac{\hat{L}}{2}\|y - x\|_2^2 + \frac{\tau\epsilon}{2}$ **then** $\hat{L} := 2\hat{L}$ **end if**
   **until** $\phi(y) \leq \phi(x) + \langle\nabla\phi(x), y - x\rangle + \frac{\hat{L}}{2}\|y - x\|_2^2 + \frac{\tau\epsilon}{2}$
   Set

$$
\begin{aligned}
x_{t+1} := x, \quad y_{t+1} := y, \quad a_{t+1} := a, \\
A_{t+1} := A_t + a_{t+1}, \quad \hat{L} := \hat{L}/2,
\end{aligned}
$$

**end for**
**Output :** $x = y_T$

---

---

**Algorithm 5** Gradient descent method

---
**Inputs :** $x_0, L_0$
**Initialize :** $\hat{L} := L_0$
**for** $t = 0, \ldots$ **do**
    **repeat**
        $x := \arg\min_z \nabla\phi(x)^T z + g(z) + \hat{L}D_h(z; x)$
        **if** $\phi(x) \geq \phi(x_t) + \langle\nabla\phi(x_t), x - x_t\rangle + \frac{\hat{L}}{2}\|x - x_t\|_2^2$ **then** $\hat{L} = 2\hat{L}$ **end if**
    **until** $\phi(x) \leq \phi(x_t) + \langle\nabla\phi(x_t), x - x_t\rangle + \frac{\hat{L}}{2}\|x - x_t\|_2^2$
    Set
$$x_{t+1} := x, \quad \hat{L} := \hat{L}/2$$

**end for**

---

## 9.2 Accelerated gradient method

The accelerated gradient method is a special instance of the universal fast gradient method when the function $\phi$ is known to be smooth (i.e. satisfies (Generic Smooth) with $s = 2$). In that case, the optimal $\epsilon$ to run the Universal Fast Gradient method is 0 (otherwise it depends on the parameters of the function). Given an initial point $x_0$, accelerated gradient method outputs, after $t$ iterations, a point $x \triangleq \mathcal{A}(x_0, t) = \mathcal{U}(x_0, 0, t)$ such that

$$f(y) - f^* \leq \frac{cL}{t^2} D_h(x_0, X^*),$$

where $D_h(x; X^*) = \min_{x^* \in X^*} D_h(x; x^*)$ is the Bregman distance from $x$ to the set of minimizers and $c = 8$. In the Euclidean setting, $D_h(y; x) = \frac{1}{2}\|x - y\|_2^2$, such that we get the bound given in (3). Here again smoothness parameter $L$ is found by a backtracking line search such that we only need a first estimate of its value.

## 9.3 Gradient descent method

We recall in Algorithm 5 the simple gradient descent method when the function $\phi$ is smooth with constant $L$. It starts at a point $x_0$ and outputs iterates $x_t = \mathcal{G}(x_0, t)$ such that

$$f(x_t) - f^* \leq \frac{2L}{t} D_h(x_0, X^*),$$

where $D_h(x; X^*) = \min_{x^* \in X^*} D_h(x; x^*)$ is the Bregman distance from $x$ to the set of minimizers. In the Euclidean setting, $D_h(y; x) = \frac{1}{2}\|x - y\|_2^2$, such that we get the bound in (14). Once again it performs a line search on the smoothness parameter $L$ such that $L_0$ can be chosen arbitrarily.

# 10 Missing Proofs

## 10.1 Proof for adaptive scheduled restarts

To prove adaptivity with the log-scale grid search strategy we need first the following Corollary of Proposition 2.2. This also shows that scheduled restart schemes are theoretically efficient only if the algorithm itself makes a sufficient number of iterations to decrease the objective value.

**Corollary 10.1.** *Let $f$ be a smooth convex function satisfying (Smooth) with parameters $(2, L)$ and (Sharp) with parameters $(r, \mu)$ on a set $K$. Assume that we are given $x_0 \in \mathbb{R}^n$ such that $\{x : f(x) \leq f(x_0)\} \subset K$. Run Algorithm 1 from $x_0$ with general schedules of the form*

$$\begin{cases} t_k = C & \text{if } \tau = 0, \\ t_k = Ce^{\alpha k} & \text{if } \tau > 0, \end{cases}$$

*we have the following complexity bounds, if $\tau = 0$ and $C \geq C^*_{\kappa,0}$,*

$$f(\hat{x}) - f^* \leq \left(\frac{c\kappa}{C^2}\right)^{\frac{N}{C}} (f(x_0) - f^*), \tag{17}$$

*while, if $\tau > 0$ and $C \geq C(\alpha)$,*

$$f(\hat{x}) - f^* \leq \frac{f(x_0) - f^*}{(\alpha e^{-\alpha} C^{-1} N + 1)^{\frac{2}{\tau}}}, \tag{18}$$

*where*

$$C(\alpha) \triangleq e^{\frac{\alpha(1-\tau)}{\tau}} (c\kappa)^{\frac{1}{2}} (f(x_0) - f^*)^{-\frac{\tau}{2}}, \tag{19}$$

*and $N = \sum_{k=1}^{R} t_k$ is the total number of iterations.*

*Proof.* Given general schedules of the form

$$\begin{cases} t_k = C & \text{if } \tau = 0, \\ t_k = Ce^{\alpha k} & \text{if } \tau > 0, \end{cases}$$

the best value of $\gamma$ satisfying condition (9) for any $k \geq 0$ in Proposition 2.2 are given by

$$\begin{cases} \gamma = \log\left(\frac{C^2}{c\kappa}\right) & \text{if } \tau = 0 \text{ and } C \geq C_{\kappa,0}^*, \\ \gamma = \frac{2\alpha}{\tau} & \text{if } \tau > 0 \text{ and } C \geq C(\alpha). \end{cases}$$

As in Proposition 2.2, plugging these values into the bounds of Lemma 2.1 yields the desired result.
∎

We can now prove Proposition 2.3 that we recall here. Notations are the same as in Proposition 2.2

**Proposition.** *Let $f$ be a smooth convex function satisfying (Smooth) with parameters $(2, L)$ and (Sharp) with parameters $(r, \mu)$ on a set $K$. Assume that we are given $x_0 \in \mathbb{R}^n$ such that $\{x \mid f(x) \leq f(x_0)\} \subset K$ and denote $N$ a given number of iterations. Run schemes $\mathcal{S}_{i,j}$ defined in (11) to solve (P) for $i \in [1, \ldots, \lfloor \log_2 N \rfloor]$ and $j \in [0, \ldots, \lceil \log_2 N \rceil]$, stopping each time after $N$ total inner algorithm iterations i.e. for $R$ such that $\sum_{k=1}^{R} t_k \geq N$.*

*Assume $N$ is large enough, so $N \geq 2C_{\kappa,\tau}^*$, and if $\frac{1}{N} > \tau > 0$, $C_{\kappa,\tau}^* > 1$.*

*If $\tau = 0$, there exists $i \in [1, \ldots, \lfloor \log_2 N \rfloor]$ such that scheme $\mathcal{S}_{i,0}$ achieves a precision given by*

$$f(\hat{x}) - f^* \leq \exp\left(-e^{-1}(c\kappa)^{-\frac{1}{2}} N\right)(f(x_0) - f^*).$$

*If $\tau > 0$, there exist $i \in [1, \ldots, \lfloor \log_2 N \rfloor]$ and $j \in [1, \ldots, \lceil \log_2 N \rceil]$ such that scheme $\mathcal{S}_{i,j}$ achieves a precision given by*

$$f(\hat{x}) - f^* \leq \frac{f(x_0) - f^*}{\left(\tau e^{-1}(c\kappa)^{-\frac{1}{2}} (f(x_0) - f^*)^{\frac{\tau}{2}} (N-1)/4 + 1\right)^{\frac{2}{\tau}}}.$$

*Overall, running the logarithmic grid search has a complexity $(\log_2 N)^2$ times higher than running $N$ iterations using the optimal (oracle) scheme.*

*Proof.* Denote $N' = \sum_{k=1}^{R} t_k \geq N$ the number of iterations of a scheme $\mathcal{S}_{i,j}$. We necessarily have $N' \leq 2N$ for our choice of $C_i$ and $\tau_j$. Hence the cost of running all methods is of the order $(\log_2 N)^2$.

If $\tau = 0$ and $N \geq 2C_{\kappa,0}^*$, we have $i = \lceil \log_2 C_{\kappa,0}^* \rceil \leq \lfloor \log_2 N \rfloor$. Therefore $\mathcal{S}_{i,0}$ has been run and we can use bound (17) to show that the last iterate $\hat{x}$ satisfies

$$f(\hat{x}) - f^* \leq \left(\frac{c\kappa}{C_i^2}\right)^{\frac{N}{C_i}} (f(x_0) - f^*).$$

Using that $C_{\kappa,0}^* \leq C_i \leq 2C_{\kappa,0}^*$, we get

$$\begin{aligned} f(\hat{x}) - f^* &\leq \left(\frac{c\kappa}{(C_{\kappa,0}^*)^2}\right)^{\frac{N}{2C_{\kappa,0}^*}} (f(x_0) - f^*) \\ &\leq \exp\left(-e^{-1}(c\kappa)^{-\frac{1}{2}} N\right)(f(x_0) - f^*). \end{aligned}$$

If $\tau \geq \frac{1}{N}$ and $N \geq 2C^*_{\kappa,\tau}$, we have $j = \lceil -\log_2 \tau \rceil \leq \lceil \log_2 N \rceil$ and $i = \lceil \log_2 C^*_{\kappa,\tau} \rceil \leq \lfloor \log_2 N \rfloor$. Therefore scheme $\mathcal{S}_{i,j}$ has been run. As $C_i \geq C^*_{\kappa,\tau} \geq C(\tau_j)$, where $C(\tau_j)$ is defined in (19), we can use bound (18) to show that the last iterate $\hat{x}$ of scheme $\mathcal{S}_{i,j}$ satisfies

$$f(\hat{x}) - f^* \leq \frac{f(x_0) - f^*}{\left(\tau_j e^{-\tau_j} C_i^{-1} N + 1\right)^{\frac{2}{\tau}}}.$$

Finally, by definition of $i$ and $j$, $2\tau_j \geq \tau$ and $C_i \leq 2C^*_{\kappa,\tau}$, so

$$
\begin{aligned}
f(\hat{x}) - f^* &\leq \frac{f(x_0) - f^*}{\left(\tau e^{-\tau_j} (C^*_{\kappa,\tau})^{-1} N/4 + 1\right)^{\frac{2}{\tau}}} \\
&= \frac{f(x_0) - f^*}{\left(\tau e^{-1} (c\kappa)^{-\frac{1}{2}} (f(x_0) - f^*)^{\frac{\tau}{2}} N/4 + 1\right)^{\frac{2}{\tau}}},
\end{aligned}
$$

where we concluded by expanding $C^*_{\kappa,\tau} = e^{1-\tau} (c\kappa)^{\frac{1}{2}} (f(x_0) - f^*)^{-\frac{\tau}{2}}$ and using that $\tau \geq \tau_j$.

If $\frac{1}{N} > \tau > 0$ and $N > 2C^*_{\kappa,\tau}$, we have $i = \lceil \log_2 C^*_{\kappa,\tau} \rceil \leq \lfloor \log_2 N \rfloor$, so scheme $\mathcal{S}_{i,0}$ has been run. Its iterates $x_k$ satisfy, with $1 - \tau = 2/r$,

$$
\begin{aligned}
f(x_k) - f^* &\leq \frac{c\kappa}{C_i^2} (f(x_{k-1}) - f^*)^{\frac{2}{r}} \\
&\leq \left(\frac{c\kappa}{C_i^2}\right)^{\left(1-(1-\tau)^k\right)/\tau} (f(x_0) - f^*)^{(1-\tau)^k} \\
&\leq \left(\frac{c\kappa (f(x_0) - f^*)^{-\tau}}{C_i^2}\right)^{\left(1-(1-\tau)^k\right)/\tau} (f(x_0) - f^*).
\end{aligned}
$$

Now $C_i \geq C^*_{\kappa,\tau} = e^{1-\tau} (c\kappa)^{\frac{1}{2}} (f(x_0) - f^*)^{-\frac{\tau}{2}}$ and $C_i R \geq N$, therefore last iterate $\hat{x}$ satisfies

$$f(\hat{x}) - f^* \leq \exp\left(-2(1 - \tau) \frac{1 - (1-\tau)^{N/C_i}}{\tau}\right) (f(x_0) - f^*).$$

As $N \geq C_i$, since

$$h(\tau) = \frac{(1 - \tau)\left(1 - (1-\tau)^{\frac{N}{C_i}}\right)}{1 - (1-\tau)}$$

is decreasing with $\tau$ and $\frac{1}{N} > \tau > 0$, we have

$$
\begin{aligned}
f(\hat{x}) - f^* &\leq \exp\left(-2(N-1)\left(1 - \left(1 - \frac{1}{N}\right)^{N/C_i}\right)\right) (f(x_0) - f^*) \\
&\leq \exp\left(-2(N-1)\left(1 - \exp\left(-\frac{1}{C_i}\right)\right)\right) (f(x_0) - f^*) \\
&\leq \exp\left(-2\frac{N-1}{C_i}\left(1 - \frac{1}{2C_i}\right)\right) (f(x_0) - f^*).
\end{aligned}
$$

having used the facts that $(1 + ax)^{\frac{b}{x}} \leq \exp(ab)$ if $ax \geq -1$, $\frac{b}{x} \geq 0$ and $1 - x + \frac{x^2}{2} \geq \exp(-x)$ when $x \geq 0$. By assumption $C^*_{\kappa,\tau} \geq 1$, so $C_i \geq 1$ and finally

$$
\begin{aligned}
f(\hat{x}) - f^* &\leq \exp\left(-\frac{N-1}{C_i}\right) (f(x_0) - f^*) \\
&\leq \exp\left(-\frac{N-1}{2C^*_{\kappa,\tau}}\right) (f(x_0) - f^*) \\
&\leq \frac{f(x_0) - f^*}{\left(\tau (C^*_{\kappa,\tau})^{-1} (N-1)/4 + 1\right)^{\frac{2}{\tau}}} \\
&\leq \frac{f(x_0) - f^*}{\left(\tau (f(x_0) - f^*)^{\frac{\tau}{2}} e^{-1} (c\kappa)^{-\frac{1}{2}} (N-1)/4 + 1\right)^{\frac{2}{\tau}}}.
\end{aligned}
$$

using the fact that $e^\tau \geq 1$. ∎

## 10.2 Proof for universal scheduled restarts

**Proposition.** *Let $f$ be a convex function satisfying* (Smooth) *with parameters $(s, L)$ on a set $J$ and* (Sharp) *with parameters $(r, \mu)$ on a set $K$. Given $x_0 \in \mathbb{R}^n$ assume that $\{x | f(x) \leq f(x_0)\} \subset J \cap K$. Run Algorithm 2 from $x_0$ for a given $\epsilon_0 \geq f(x_0) - f^*$ with*

$$\gamma = \rho, \qquad t_k = C^*_{\kappa,\tau,\rho} e^{\tau k}, \quad \text{where} \quad C^*_{\kappa,\tau,\rho} \triangleq e^{1-\tau}(c\kappa)^{\frac{s}{2\rho}} \epsilon_0^{-\frac{\tau}{\rho}}$$

*where $\rho$ is defined in* (12)*, $\kappa$ and $\tau$ are defined in* (2) *and $c = 8e^{2/e}$ here. The precision reached at the last point $\hat{x}$ is given by,*

$$f(\hat{x}) - f^* \leq \exp\left(-\rho e^{-1}(c\kappa)^{-\frac{s}{2\rho}} N\right) \epsilon_0 = O\left(\exp(-\kappa^{-\frac{s}{2\rho}} N)\right), \quad \text{when } \tau = 0,$$

*while,*

$$f(\hat{x}) - f^* \leq \frac{\epsilon_0}{\left(\tau e^{-1}(c\kappa)^{-\frac{s}{2\rho}} \epsilon_0^{\frac{\rho}{s}} N + 1\right)^{-\frac{\rho}{\tau}}} = O\left(\kappa^{\frac{s}{2\tau}} N^{-\frac{\rho}{\tau}}\right), \quad \text{when } \tau > 0,$$

*where $N = \sum_{k=1}^R t_k$ is total number of iterations.*

*Proof.* Our goal is to ensure that the target accuracy is reached at each restart, i.e.

$$f(x_k) - f^* \leq \epsilon_k. \tag{20}$$

By assumption, (20) holds for $k = 0$. Assume that (20) is true at iteration $k - 1$, combining (Sharp) with the complexity bound in (13), then

$$\begin{aligned}
f(x_k) - f^* &\leq \frac{\epsilon_k}{2} + \frac{c\kappa(f(x_{k-1}) - f^*)^{\frac{2}{r}}}{\epsilon_k^{\frac{2}{s}} t_k^{\frac{2\rho}{s}}} \frac{\epsilon_k}{2} \\
&\leq \frac{\epsilon_k}{2} + \frac{c\kappa}{t_k^{\frac{2\rho}{s}}} \frac{\epsilon_{k-1}^{\frac{2}{r}}}{\epsilon_k^{\frac{2}{s}}} \frac{\epsilon_k}{2},
\end{aligned}$$

where $c = 8e^{2/e}$ using that $r^{2/r} \leq e^{2/e}$. By definition $\epsilon_k = e^{-\gamma k}\epsilon_0$, so to ensure (20) at iteration $k$ this imposes

$$\frac{c\kappa e^{\gamma\frac{2}{r}} e^{-\gamma\left(\frac{2}{r} - \frac{2}{s}\right)k}}{t_k^{\frac{2\rho}{s}}} \epsilon_0^{\frac{2}{r} - \frac{2}{s}} \leq 1.$$

Rearranging terms in last inequality, using $\tau$ defined in (2),

$$t_k \geq e^{\gamma\frac{1-\tau}{\rho}}(c\kappa)^{\frac{s}{2\rho}} \epsilon_0^{-\frac{\tau}{\rho}} e^{\frac{\gamma\tau}{\rho}k}.$$

Choosing $t_k = Ce^{\alpha k}$, where

$$C = e^{\gamma\frac{1-\tau}{\rho}}(c\kappa)^{\frac{s}{2\rho}} \epsilon_0^{-\frac{\tau}{\rho}} \qquad \text{and} \qquad \alpha = \frac{\gamma\tau}{\rho},$$

and using Lemma 2.1 then yields,

$$f(\hat{x}) - f^* \leq \exp(-\gamma e^{-\frac{\gamma}{\rho}}(c\kappa)^{-\frac{s}{2\rho}} N)\epsilon_0, \tag{21}$$

when $\tau = 0$, while,

$$f(\hat{x}) - f^* \leq \frac{\epsilon_0}{\left(\frac{\gamma\tau}{\rho} e^{-\frac{\gamma}{\rho}}(c\kappa)^{-\frac{s}{2\rho}} \epsilon_0^{\frac{\tau}{\rho}} N + 1\right)^{\frac{\rho}{\tau}}}. \tag{22}$$

when $\tau > 0$. These bounds are minimal for $\gamma = \rho$ and the results follow. ∎

## 10.3 Proof for restarts with termination criterion

**Proposition.** *Let $f$ be a convex function satisfying (Smooth) with parameters $(s, L)$ on a set $J$ and (Sharp) with parameters $(r, \mu)$ on a set $K$. Given $x_0 \in \mathbb{R}^n$ assume that $\{x, \ f(x) \leq f(x_0)\} \subset J \cap K$. Run Algorithm 3 from $x_0$ with parameter $\gamma = \rho$. The precision reached at the point $x_R$ is given by,*

$$f(\hat{x}) - f^* \ \leq \ \exp\left(-\rho e^{-1}(c\kappa)^{-\frac{s}{2\rho}} N\right)(f(x_0) - f^*) \ = \ O\left(\exp(-\kappa^{-\frac{s}{2\rho}} N)\right), \quad \text{when } \tau = 0,$$

*while,*

$$f(\hat{x}) - f^* \ \leq \ \frac{f(x_0) - f^*}{\left(\tau e^{-1}(c\kappa)^{-\frac{s}{2\rho}}(f(x_0) - f^*)^{\frac{\tau}{\rho}} N + 1\right)^{\frac{\rho}{\tau}}} \ = \ O\left(\kappa^{\frac{s}{2\tau}} N^{-\frac{\rho}{\tau}}\right), \quad \text{when } \tau > 0,$$

*where $N$ is the total number of iterations, $\rho$ is defined in (12), $\kappa$ and $\tau$ are defined in (2) and $c = 8e^{2/e}$ here.*

*Proof.* Given $\gamma \geq 0$, linear convergence of our scheme is ensured by our choice of target accuracies $\epsilon_k$. It remains to compute the number of iterations $t_{\epsilon_k}$ needed by the algorithm before the $k^{\text{th}}$ restart. Following proof of Proposition 3.1, for $k \geq 1$ we know that target accuracy is necessarily reached after

$$\bar{t}_k = e^{\gamma \frac{1-\tau}{\rho}}(c\kappa)^{\frac{s}{2\rho}}\epsilon_0^{-\frac{\tau}{\rho}} e^{\frac{\gamma\tau}{\rho}k}$$

iterations, such that $t_{\epsilon_k} \leq \bar{t}_k$. So Algorithm 3 achieves linear convergence while needing less inner iterates than the scheduled restart presented in Proposition 3.1, its convergence is therefore at least as good. For a given $\gamma$ bounds (21) and (22) follow with $\epsilon_0 = f(x_0) - f^*$ and taking $\gamma = \rho$ is optimal. ∎