[Reviews · NeurIPS 2017]

Reviewer 1



This paper consider first-order algorithms for Holder-smooth convex optimization in the oracle model with an additional sharpness assumption, guaranteeing that, within a neighborhood of optimum, a reduction in objective value yields a reduction in distance from optimum. Recently, there has been growing interest in the algorithmic consequences of the presence of sharpness, particularly in the setting of alternating minimization and of compressed sensing. Sharpness can be exploited to speed up the convergence of first-order methods, such as Nesterov's accelerated gradient descent, by appropriately restarting the algorithm after a certain number of iterations, possibly changing with the number of rounds. First, the authors provide asymptotically optimal restart schedules for this class of problem for given sharpness parameters mu and r. While this is interesting, the result is essentially the same as that appearing, in more obscure terms, in Nemirovski and Nesterov's original 1985 paper "Optimal methods of smooth convex optimization". See paragraph 5 of that paper. More importantly, the authors show that a log-scale grid search can be performed to construct adaptive methods that work in settings when mu and r are unknown, which is typical in sharpness applications. This appears to be the main novel idea of the paper. From a theoretical point of view, I find this is to be a fairly straightforward observation. On the other hand, such observation may be important in practice. Indeed, the authors also show a small number of practical examples in the context of classification, in which the restart schedules significantly improve performance. At the same time, the fact that restarts can greatly help the convergence of accelerated methods has already been observed before (see O'Donoghue and Candes, as cited in the paper). In conclusion, I find the paper interesting from a practical point of view and I wish that the authors had focused more on the empirical comparison of their restart schedule vs that of Nemirovski and Nesterov and others. From a theoretical point of view, my feeling is that the contribution is good but probably not good enough for NIPS. It might help if the authors, in their rebuttal, explained more clearly the relation of their non-adaptive bounds with those of Nemirovski and Nesterov.

Reviewer 2



Summary of the paper ==================== This paper considers restarting schemes which allow one to explicitly incorporate growth properties (namely, sharpness) of convex functions into algorithms which do not necessarily exploit this additional favorable assumptions. First, the number of inner iterations per epoch is scheduled based on the parameters of the growth condition. As these parameters are hard to approximate, an adaptive scheduling is devised based on parameters grid search. Finally, it is shown that one can obtain a near-optimal rate only by knowing the value of the minimizer (omitting the requirement for knowing the sharpness parameters). Evaluation ========== The main contribution of of the paper is combining the mechanism of restarting schemes with the growth conditions of convex functions. The actual rate obtained by this technique seem to be of a somewhat narrow practical value (requires strong prior knowledge or grid search). However, from theoretical standpoint, it is an interesting general approach of exploiting sharpness. That said, the paper seems to contribute to the study of restarting mechanisms schemes only incrementally. The paper is well-written and easy to follow. General Comments ================ - A more through discussion regarding other existing algorithms which obtain the same optimal rate is missing. - Related Work which may be worth mentioning: - similar upper bound: https://arxiv.org/pdf/1609.07358.pdf - lower bound using restarting scheme http://proceedings.mlr.press/v48/arjevani16.pdf Minor Comments ============== - L16: Might worth emphasizing that f^* is taken over K (and not, e.g., the domain over which f is defined). - L153: In what sense should we expect convergence? - L217: Didn't find the definition for Q before this line (I did find a definition in the next section). - L285 (appendix): broken reference ??